# Influences on Apartment Design: A History of the Spatial Layout of Apartment Buildings in Sydney and Implications for the Future

**Hyungmo Yang** [1,*] , **Philip Oldfield** [1] **and Hazel Easthope** [2]

1 School of Built Environment, Faculty of Arts, Design & Architecture, University of New South Wales, Sydney, NSW 2052, Australia; p.oldfield@unsw.edu.au
2 City Futures Research Centre, Faculty of Arts, Design & Architecture, University of New South Wales, Sydney, NSW 2052, Australia; hazel.easthope@unsw.edu.au
* Correspondence: hyungmo.yang@unsw.edu.au or yanghyungmo@gmail.com

**Abstract:** This paper traces the history of apartment design with an emphasis on spatial layout. It charts the events that have influenced apartment design in Sydney, Australia and provides a framework for understanding how changes in society, the economy, regulations, and architectural paradigms have influenced apartment layouts over time. Through a review of historical and contemporary apartment plan drawings in Sydney, we identify four chronologically distinct eras: layouts reflecting physically separate rooms and a healthier living condition (1900–1935); layouts following function (1935–1961); layouts enhancing interaction between family members (1961–2002); and layouts for independent life and to satisfy minimum regulatory requirements (2002–the present). We then consider these distinct eras in relation to political, economic, and social influences at the time. We propose that prior to 1961, changes in social paradigms and architectural thinking and the development of technologies were the main drivers of apartment layouts. After 1961, changes in the economy, the housing market, and regulations appear to have had more influence. This historical analysis provides insights into factors contributing to current apartment layouts and how different social, economic, and regulatory levers may influence them in future. These insights will be useful to both practitioners and academics in international jurisdictions considering how to encourage improved apartment spatial layouts in future.

**Keywords:** apartment buildings; spatial layouts; influences; architectural history; Sydney

## 1. Introduction

Globally, more than half of the world's population lives in urban areas, and this proportion is projected to increase to around 70 per cent by 2050 [1]. Rapid population growth and urbanisation have resulted in apartments becoming a common housing type in urban areas all over the world. However, despite its proliferation, there remains significant criticism of current apartment building and unit design and its appropriateness for some demographics, especially families with children [2–5]. Internationally, researchers and practitioners such as architects, planners, and regulators are endeavouring to improve the quality and experience of apartment living. In particular, some researchers [6,7] have found that the spatial layout is an important attribute influencing the quality of apartment living.

In the field of architectural studies, there have been numerous historical approaches to analysing the spatial layout of apartment buildings. For example, Alitajer and Nojoumi [8] compared the layout of traditional and modern apartment buildings. Other studies [9–14] have explored the features of spatial layouts in apartment buildings chronologically. Some research [15,16] has focused on specific rooms, investigating the periodic characteristics of, for example, kitchens and corridors within certain countries. These studies describe how

apartment building layouts have changed over time. However, there are few studies (except [9,13,14]) that examine the influence of external factors on spatial layouts of apartment buildings over time. Given the findings of previous research that apartment layouts can be influenced by societal factors [13,14] and the economy [9], exploring historical connections between spatial layout and external influences can assist in a deeper understanding of why apartment building layouts are the way they are. Thus, this paper examines the influence of social and economic factors on apartment design and spatial layout in particular. We also consider changes in regulation and architectural technologies. We expect that these insights will be useful to both practitioners and academics in international jurisdictions who are considering how to encourage improved apartment layouts in the future.

This paper focuses on apartment layouts in Sydney, Australia. Despite a history of over a century of apartment building in Sydney, few academic studies on the design of apartment buildings in the city have been undertaken ([17–19] being notable exceptions). This might be explained by the dominance of traditional detached housing in Sydney and Australia in general. However, there has been a notable increase of apartment buildings approved in Sydney [20], as well as an increasing population calling these buildings home [21–26]. By the early 2000s, around half of new residential building approvals in New South Wales (NSW) were attached dwellings (including apartments and townhouses), with the proportion rising further in the period 2015–2018 (Figure 1). While the proportion of attached dwellings has dipped in recent years, it still remains high, accounting for almost half of all dwelling completions (Figure 1). In Sydney, around 20 per cent of residents (247,818 persons) lived in apartments in 1991, increasing to around 30 per cent (456,233 persons) by 2016 (Figure 2). It is an appropriate time to look back on the history of apartment building designs while simultaneously considering their future.

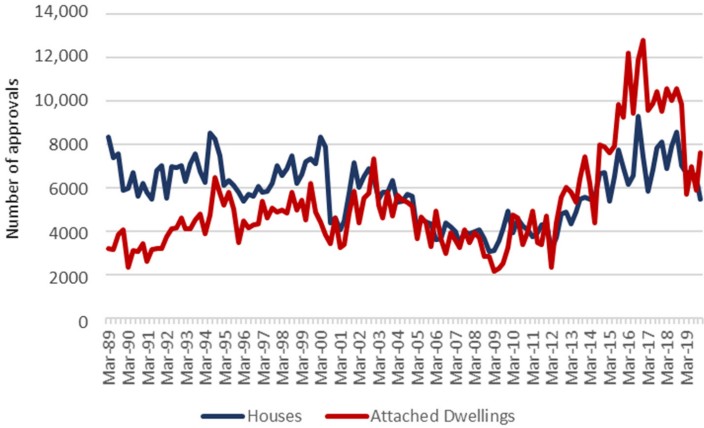

**Figure 1.** Dwelling starts, NSW (Adapted from Ref. [27]).

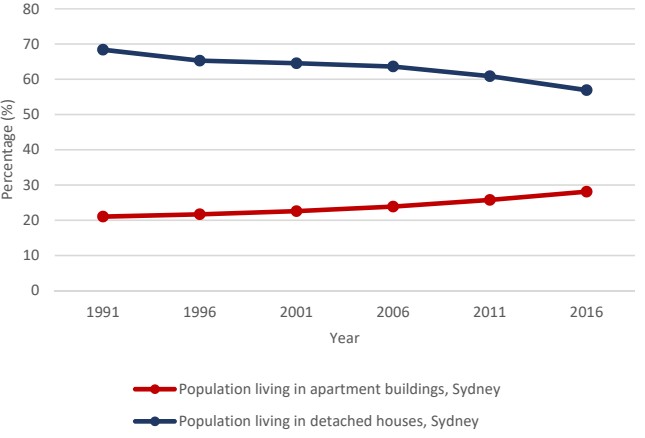

**Figure 2.** Population living in detached houses and apartment buildings in Sydney (Adapted from Refs. [21–26]).

## 2. Materials and Methods

This paper investigates how the spatial layout in apartment buildings and units in Sydney has been influenced by historical events through the literature and plan drawing reviews. The paper defines spatial layout as the composition of spaces, their position in the apartment buildings and units, and the relationship between different spaces. This is a different approach to Butler-Bowden's notable study on the history of apartments in Sydney, which is concerned with "the changes in apartment types (walk-up, slab, tower), locations and governance against a background of debates for and against apartments" (for more detail, see [17,19]).

The literature review focused on (1) the features of spatial layout and (2) the events which may have influenced layout designs. For the historical analysis, books, magazines, newspapers, dissertations, and academic articles on housing design, including apartment buildings in Sydney, were reviewed. The plan drawing review was based on apartment buildings constructed in Sydney from 1900 to the present day. The cases have been collected from books and architectural magazines such as "Architecture Australia" and "Building: the magazine for the architect, builder, property owner and merchant". In total, 44 apartment building plans were analysed as part of this study, with 11 referenced in the paper as case studies to illustrate specific characteristics of spatial layout (two to four plans per era in the following sections). The cases selected do not necessarily represent all types of spatial layouts in apartment buildings and units at the time. However, the comparative analysis of the spatial layout of floor plans reviewed demonstrates historically distinct features which can be categorised according to chronological eras. Therefore, the plan drawing analysis assisted in verifying the descriptive features of apartment layouts in the written literature and in exploring the links between apartment building design and historical events.

Using this research, we have categorised four 'eras' of apartments in Sydney, grouped by common characteristics in terms of their spatial layout and through the influence of economic, social, and architectural factors of the time. The specific dates chosen to define the eras are based on significant historical events (in a manner similar to [28]). The grouping of architectural typologies by era is a common approach, with other examples including categorising buildings chronologically by their environmental characteristics [28], architectural style [29] or technical design [30].

## 3. Analysis

Through the literature review, we have selected key events that have influenced the spatial layouts at the time:

1.  The beginning of conservatism and the outbreak of plague (starting in 1900);
2.  The growth of modernism (starting in 1935) and developments post World War II (Note: The house built at Castlecrag (completed in 1935), designed by Walter Burley Griffin and Marion Mahony, is considered the first example of modernist housing in Sydney [31]);
3.  Postwar population growth, growth in the marriage rate, and the introduction of strata title (1961) (Note: The population in Australia escalated in this era because of the postwar "baby boom" from 1946 and the increase in migration between 1954 and 1961 [19]);
4.  Changes in demographic characteristics of households living in apartment buildings and the introduction of apartment design regulation (2002) (Note: The state of New South Wales, Australia, enacted "State Environmental Planning Policy No 65 (SEPP 65)—Design Quality of Residential Apartment Development" to inform apartment design and performance as of 2002.)

Through the methods outlined above, four eras of apartment building layouts in Sydney are identified:

1.  1900–1935, Layouts reflecting physically separate rooms and healthier living conditions;
2.  1935–1961, Layouts following function;

3.   1961–2002, Layouts enhancing interaction between family members;
4.   2002–the present, Layouts for independent life and satisfaction of minimum regulatory requirements.

These factors are explored in more depth in the following sections.

### 3.1. Apartment Layouts Reflecting Physically Separate Rooms and a Healthier Living Condition: 1900–1935

Sydney's first apartment buildings were constructed in the early 20th century. Initially, apartment buildings were built mostly for the wealthy, emphasising the convenience of living, and influenced by a servant shortage [19]. Sydney also faced the same problems as many cities at the time. This included the outbreak of bubonic plague as well as crowding and unsanitary living environments [19]. In line with this, in 1912, Robert Irvine, a professor at Sydney University, carried out a project for the NSW government to examine dwellings in the USA and Europe and recommended the promotion of specific housing design characteristics, including well-ventilated apartment buildings [17,19]. This project provided an opportunity for the NSW government architects to consider international trends in apartment building design at the time, such as the integration of lightwells for natural light and air and the inclusion of communal spaces for sanitary facilities.

A feature of unit layouts at the time was that individual functions and spaces were separated. Internal spaces such as living rooms, bedrooms, dining rooms, bathrooms, and kitchens were enclosed with physical walls and were accessible only through an entrance or opening from an inner hall or corridor. This segregation of function was also apparent in earlier housing types [32] and is connected to both social and safety considerations. For instance, social norms amongst the middle classes encouraged the separation of kitchens from other living spaces [33]. The desire for fire safety also influenced the separation of the kitchen from other inhabitable rooms [34].

As a result of these physically separate spaces, lobbies or corridors in individual units played an important role, not only as a buffer zone between different rooms but also as a central space providing access to the various rooms in a unit (Figure 3).

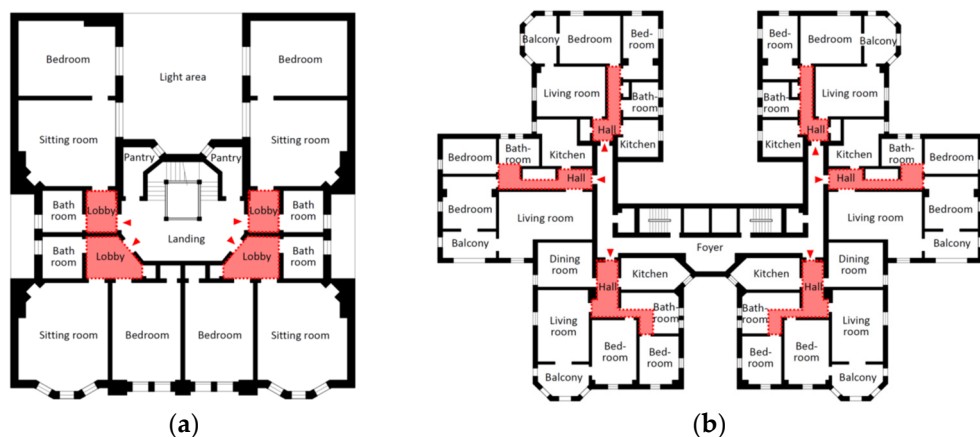

(a)                                    (b)

**Figure 3.** Centralised lobby or corridor in a unit through which most rooms were accessible. (**a**) Strathkyle, Sydney, 1909 (This example is likely an apartment hotel with shared kitchen and dining rooms) (Adapted from Ref. [17]); (**b**) Borambil, Manly, 1929 (Adapted from Ref. [17]).

Many units at this time had open fireplaces in living rooms and bedrooms. Open fireplaces had the functions of heating and lighting as well as sometimes being used for cooking [35]. Open fireplaces were normally set into the wall with a chimney for smoke exhaust. From a functional perspective, this meant that living rooms and bedrooms were often organised opposite each other, allowing for a shared party wall and thus a shared chimney flue (Figure 4). As a result, the zoning of a living room, a bedroom, and an open fireplace was a feature of units in this era.

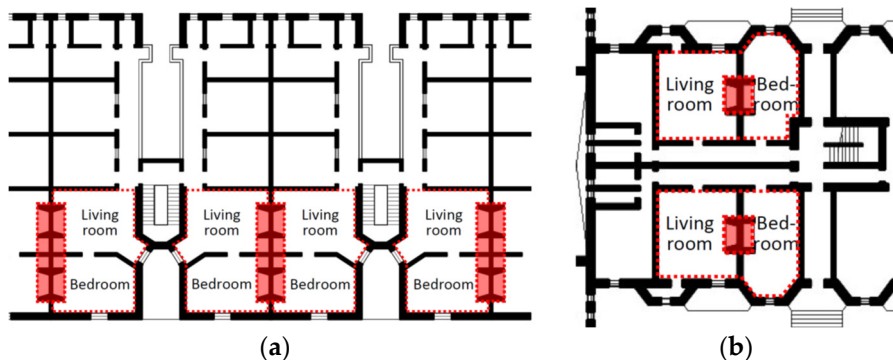

**Figure 4.** Planning of living and bedroom spaces to allow for shared chimney flues. (**a**) Millers point project, Sydney, 1910 (Adapted from Ref. [36]); (**b**) Chippendale project, Sydney, 1914 (Adapted from Ref. [36]).

Another common feature of unit layouts was the main bedroom which directly (or indirectly through a balcony) faced the street (Figure 5). This enabled residents to overlook and monitor the outer street from the main bedroom. The ability for residents to monitor the street was an important quality in apartment buildings in this era. For example, according to an article in a weekly newspaper at the time [37], Stevens' Tenement Buildings (built in 1900) was consciously designed "*having its own lookout windows in front and rear*.

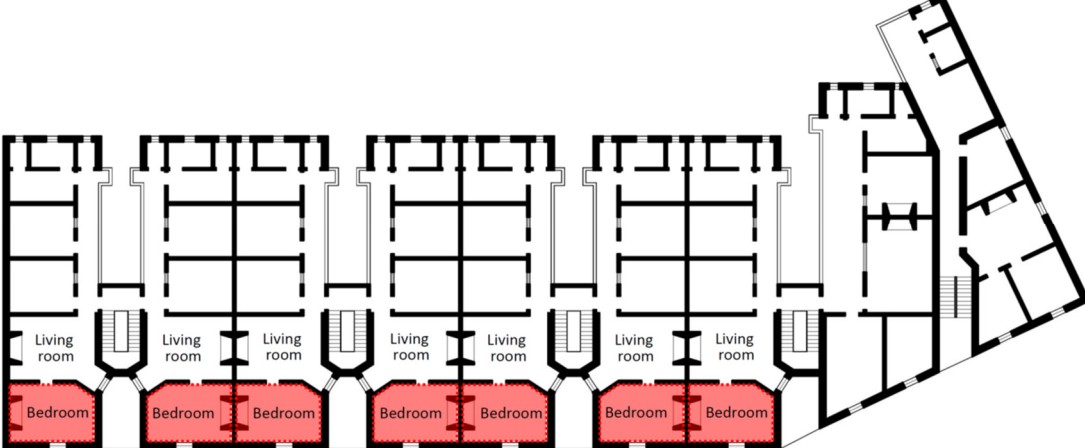

**Figure 5.** Bedrooms facing the main street, with living spaces behind. Millers point project, Sydney, 1910 (Adapted from Ref. [36]).

Turning from the unit layout to the building layout, communal spaces for sanitary facilities such as a common laundry, bathroom, and sometimes kitchen areas were necessary as units themselves typically did not have all sanitary facilities self-contained in this era [17]. Also of note, the ground floor units of some buildings were accessed directly from street level [18] without a buffer zone such as a common hallway.

In terms of apartment planning, many buildings were organised in a mirrored plan on either side of the central core [19] and included lightwells to provide natural light and ventilation to improve comfort and interior conditions (Figure 6) [18,19]. Such an approach was not limited to Sydney at this time, with "Quarter block" office buildings in Chicago also including multistorey lightwells to provide interior spaces with access to light and ventilation [38]. The inclusion of lightwells and the creation of undulating plan layouts ensuring habitable rooms have access to windows is a design strategy to overcome the limitations of technologies of the era. In the early twentieth century, artificial lighting was inefficient, and while some artificial cooling and air-conditioning systems were available, they were far too expensive for mass application [39]. As such, windows

provided the primary access to light and fresh air and thus were vital for health and well-being. Such an approach is also a reflection of the aspirations of early modernism emerging as an architectural movement during the latter part of this era, which sought to improve occupant health and tackle diseases such as tuberculosis through the provision of sunlight and fresh air in building design [40]. In Sydney, recognition of the importance of health was strengthened by epidemics at the beginning of this era and influenced by regulations such as Ordinance 70 and Ordinance 71 [19,41] that established housing standards for improving the quality of living environments through natural ventilation and natural light. Ordinance 70 and 71, which partially covered the regulation for the design and construction of apartment buildings, were proclaimed in 1906 and amended in 1921, respectively. Ordinance 70, for example, set the guidance of *"a minimum floor area for rooms of 100 square feet; a minimum ceiling height for rooms other than attics, of 10 feet; at least one openable window in each room; adequate internal and under-floor ventilation; and specified damp coursing".* [41].

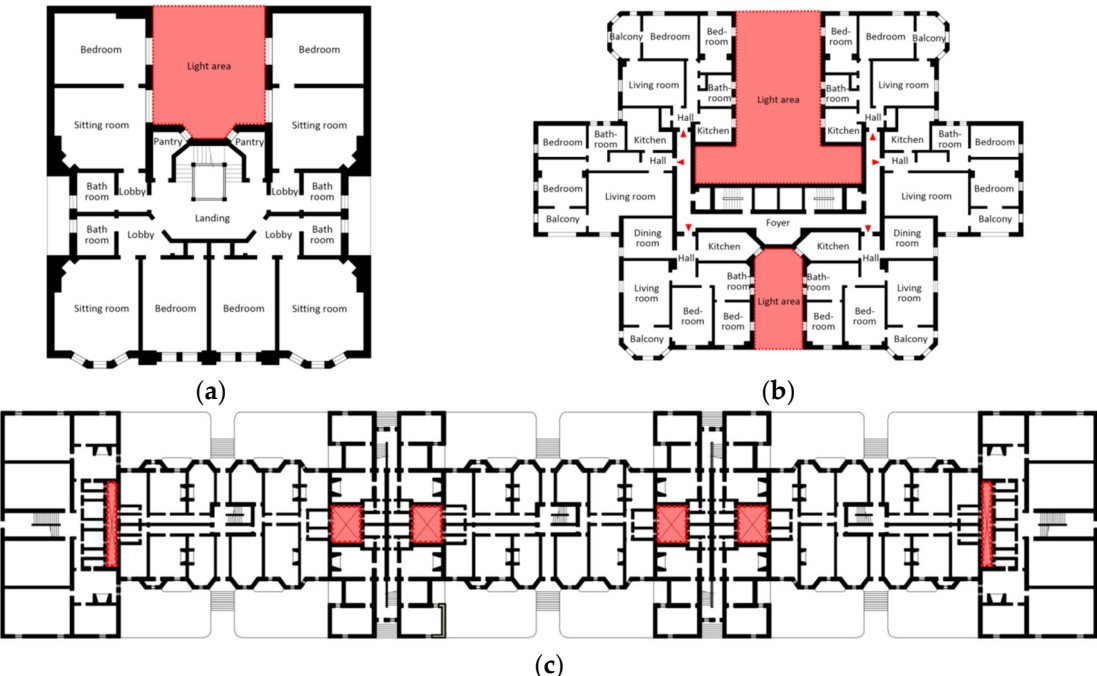

**Figure 6.** The location of lightwells in early 20th-century apartment plans to provide light and air to interior spaces. (**a**) Strathkyle, Sydney, 1909 (Adapted from Ref. [17]); (**b**) Borambil, Manly, 1929 (Adapted from Ref. [17]); (**c**) Chippendale project, Sydney, 1914 (Adapted from Ref. [36]).

### 3.2. Apartment Layouts following Function: 1935–1961

During the era 1935–1961, changes in architectural paradigms and the development of technologies had a significant influence on the spatial layouts of apartment buildings. To be specific, the growth of modernism in housing in Sydney [31] influenced apartment building design to facilitate more convenient and efficient layouts. Figure 7 shows examples of apartment buildings completed in Sydney in this era. Here the spatial layout of units was clearly divided into "serviced" and "served" zones. This could be because some of the served spaces that had been shared in the previous era (kitchen, bathroom, laundry, etc.) were no longer shared and were more commonly planned within individual units. In particular, the Wylde Street apartment building designed by Aaron M. Bolot was selected by the Australian Institute of Architects as being of significant heritage value in demonstrating modernist characteristics of design [42]. The serviced areas, such as kitchens and bathrooms, were planned into a clear zone atop one another to reduce plumbing and drainage costs [19,34].

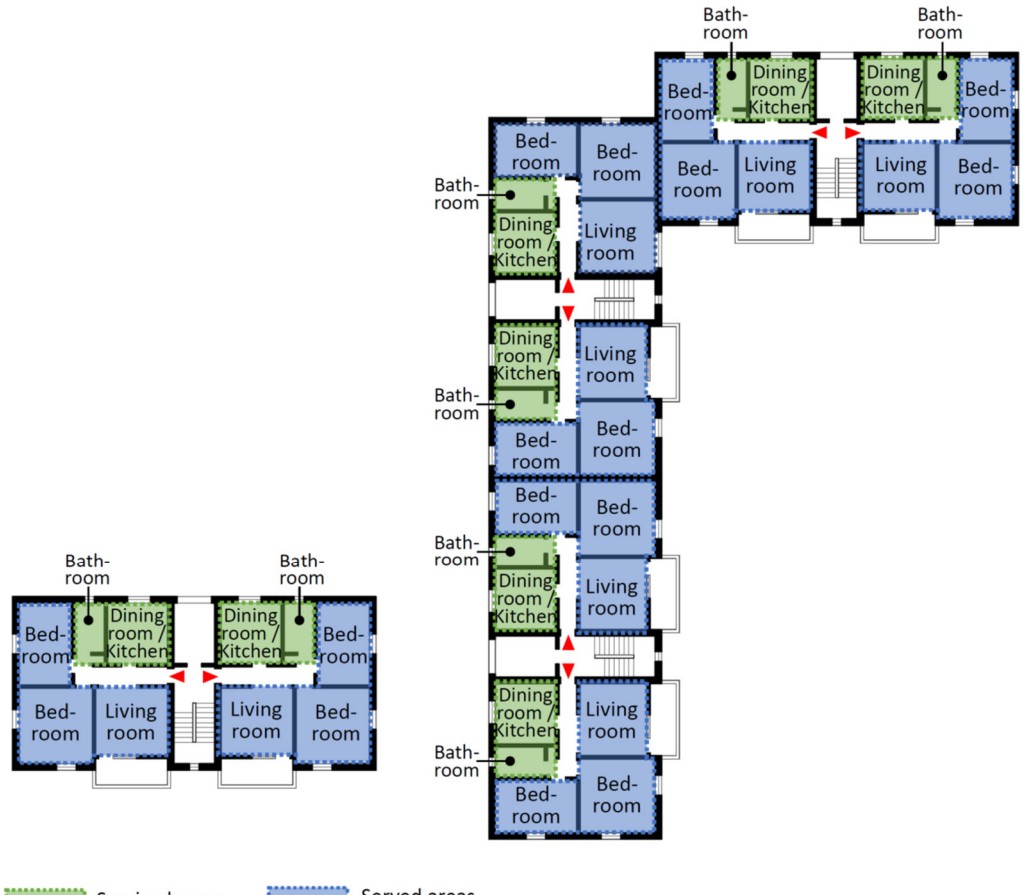

**Figure 7.** Unit layouts divided by function (the zoning of serviced and served spaces). Flats at Balmain, Sydney, 1952 (Adapted with permission from Ref. [43]).

However, physically separate spaces and a centralised hall or a corridor in individual units were still common features, similar to the previous era. For example, the kitchen in this era was often still separated from the dining room by walls with a server and a door [34]. This layout, with most rooms separated by walls, can be understood through social expectations for pleasant living conditions at the time, with a physically separate kitchen and living spaces preventing the sight of used plates, pots, and dishes after eating [34].

During this era, the design of spaces for efficient use and occupation was frequently considered. According to Supski [44], work efficiency in the kitchen with the proximity of a cooker, fridge, and a sink in a triangular layout (often indicated as an "ideal") was frequently discussed in magazines such as *Australian Home Beautiful* and *Australian House and Garden*, and in advertisements of the time. Similarly, Freeland [34] and Butler-Bowdon and Pickett [17] explained that kitchens were generally designed to minimize working movement and make working easier in this era, influenced by the modernist focus on functional design.

The development of central heating systems and boilers in this era enabled open fireplaces to be eliminated, reduced, or used only as decoration [34]. This also facilitated a change in fuel from wood and coal to gas and electricity, which have remained the dominant primary heating sources in Australia ever since [45]. Considering the layout of units had been characterised by open fireplaces adjoined to a living room and a bedroom back-to-back, in order to economically share a chimney before 1935 (see Figure 4), the elimination of open fireplaces led to the possibility of the more flexible and diverse spatial organisation of living rooms and bedrooms. As a result of these technological evolutions, some apartment buildings, such as the Wylde Street apartment in Sydney, provided kitchens with electric and gas stoves [46].

At this time, and in particular, in the postwar period, the technologies of artificial lighting and air-conditioning were becoming both more effective and more commonplace globally [28,47]. Therefore, the need for passive lighting and ventilation through lightwells became less essential to comfort and conditioning and subsequently were far less frequently used in building layouts of this era (a pattern consistent in other countries too, for example, in Croatia [48]).

This was also an era of significant events in society, demography, and economy in Sydney, which would influence the features of apartment layouts in the following years. This included the growth of population resulting from the postwar 'baby boom' and increases in migration starting in 1954 [19], which contributed to the increased demand for apartments. The rapid increase in the popularity of marriage from the 1950s [19] was also significant. According to the ABS (Australian Bureau of Statistics) [49], the percentage of the adult population who were married increased from around 55 per cent in 1933 to around 66 per cent in 1954, maintaining a similar level in the 1960s. These social and demographic changes would result in the growth rate of nuclear family households living in apartment buildings [19] and rising expectations for living environments that enabled bonds between family members in the following era (from 1961). There were also major economic shifts at this time, with the economic recovery in the 1950′s heralding a financial boom in the 1970s. These economic changes would influence the housing market in the following years, facilitating growth in the development and investment of apartment buildings in Sydney.

### 3.3. Apartment Layouts Enhancing Interaction between Family Members: 1961–2002

Because of the aforementioned societal changes in the previous era, most notably the increase in population and marriage rate resulting in an increase in consumers and the recovery of economic conditions after the war, the development and investment boom in apartment buildings in Sydney really took off from 1961 [50]. The apartment development and investment boom were accelerated in this era through the introduction of the Conveyancing (strata title) Act 1961 (NSW), which provided a legal mechanism allowing the ownership of individual strata lots (individual apartment units) [50–53], and contributed to the growth in apartment development in Sydney [54]. Before enacting the Conveyancing (strata title) Act 1961 (NSW), the majority of apartment buildings in the NSW were rented rather than owner-occupied using company title schemes. As a result, thanks to the introduction of strata title ownership, the apartment market became more diverse [19]. Easthope [54] explains that the introduction of strata title legislation in NSW "*is largely credited to a property developer, Dick Dusseldorp of Lendlease, who wanted to make individual apartments a more attractive and tradable commodity.*" Other major developers such as Meriton and Mirvac emerged shortly after, in 1963 and 1972, respectively, making their profit by purchasing land and selling apartments to individual owner/occupiers and investors [19]. Developers consequently standardised apartment building designs [19] for efficient mass production and tried to focus on the preferences of owners/occupiers and investor purchasers for successful sales.

An important change in unit layouts in this era was the emergence of open-plan layouts for shared spaces (Figure 8). Open-plan designs were made possible in this era because of the increased use of framed structures which typically used reinforced concrete beams and columns, with floors spanning between [55]. Open-plan layouts can create more efficient spaces for multifunctionality [56] as well as facilitate a more diverse use of spaces [57,58], as distinct from the monofunctional spaces of previous eras. Furthermore, the introduction of open-plan layouts in Australia in this era reflects the social expectation for home environments to facilitate easier parental supervision [59]. Given the rapid growth of population, marriage rate, and developers capitalizing on market demands through speculative development in Sydney at the time [19], open-plan layouts were an effective design strategy for delivering common units that could be flexibly occupied by different types of households in different ways. The open plan also contributed to resolving problems resulting from cramped shared spaces, particularly in smaller units.

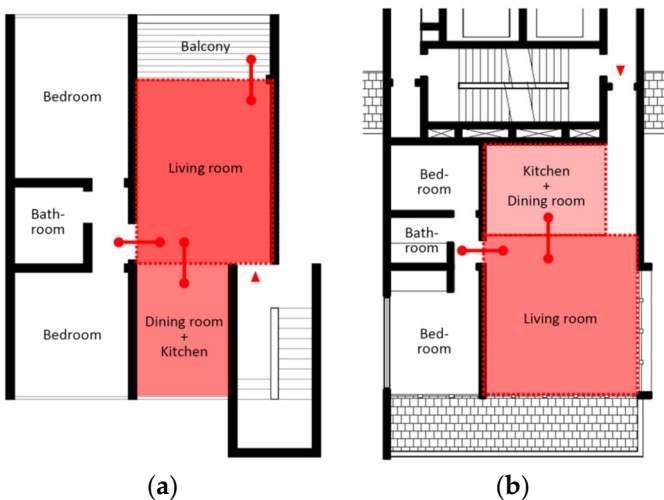

(a)                                                                                (b)

**Figure 8.** Open plan of shared spaces and a centralised living room. (**a**) Low-cost flats, Rosebery, 1967 (Adapted with permission from Ref. [60]); (**b**) Altair apartments, Rushcutters Bay, Sydney, 2001 (Adapted with permission from Ref. [61]).

In open-plan layouts, living rooms are often spatially centralised so that they must be passed through to access other spaces beyond the unit entrance (Figure 8) instead of via a centralised hall or corridor, which was common in the previous eras. These spatial characteristics reinforced the living room as the most 'public' area in a unit for family members. A living room that is spatially open onto a dining room or kitchen, as well as centrally located for circulation, enables family members to encounter each other and provides more opportunities for interaction. Considering the changes in the housing market in this era (i.e., the emergence of apartment buildings by commercial development companies), this spatial characteristic of a centralised living room in open shared spaces can be understood as a design outcome reflecting social and demographic changes at the time.

At the building scale, slab typologies demonstrated significant building layout innovation, influenced by modernist architectural precedent emerging in Europe, the U.S., and South America. Some apartments provided gallery circulation with units accessed from an open-air walkway. Examples include the Elanora Flats in North Bondi (1962), with walkways circulating around a central courtyard garden (Figure 9), and Rosebury's Housing Commission Flats (1967) with its "streets in the sky" and structurally freestanding core, characteristic typical of 'brutalist' apartment blocks at the time [62]. The particular innovation in apartment planning was brought by Viennese-born architect Harry Seidler, whose work had been highly influenced by Walter Gropius, Marcel Breuer, and Oscar Niemeyer [63]. Seidler designed a number of Sydney apartment buildings in the early part of this era with open corridor access and split-level units. Examples include Ithaca Gardens (1960), where double-height units are organized with an open-plan living space below and bedrooms above, accessed from an external gallery, or Roslyn Gardens, with east-facing open-plan living spaces and west-facing bedrooms, with a half-floor level change between [19]. This organization often meant that units were dual aspect, allowing for cross ventilation—even though domestic air-conditioning was increasingly available during this era, it was not until the 1990's that even a quarter of Australian households had access to this in their home [64].

Another notable development in apartment buildings constructed in this era was dedicated parking areas, reflecting an increase in car ownership since the 1960s [34]. Australian census data show a rapid change in vehicle ownership from one vehicle for every eight people during the 1947–1948 period, increasing to one vehicle for every four people by 1960 [65].

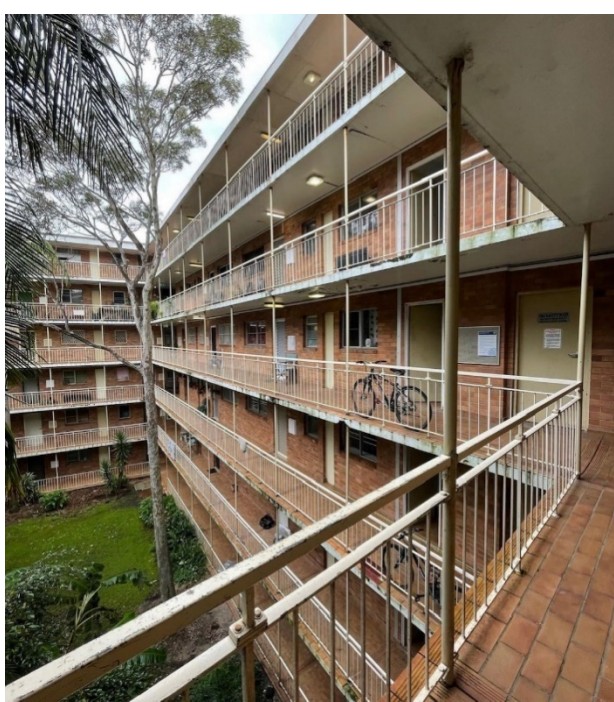

**Figure 9.** Elanora Flats, 1962, designed by Bunning and Madden with open single-loaded corridors servicing units. Source: courtesy of Philip Thalis.

*3.4. Apartment Layouts for Independent Life and to Satisfy Minimum Regulatory Requirement: 2002–The Present*

Open-plan apartment layouts continued to be the norm in this fourth era, post-2002. However, a notable change in unit layouts is the re-emergence of a centralised hall or corridor. As a result, the unit types with both a centralised hall or corridor and a centralised living room coexist in contemporary apartment buildings in Sydney (Figure 10).

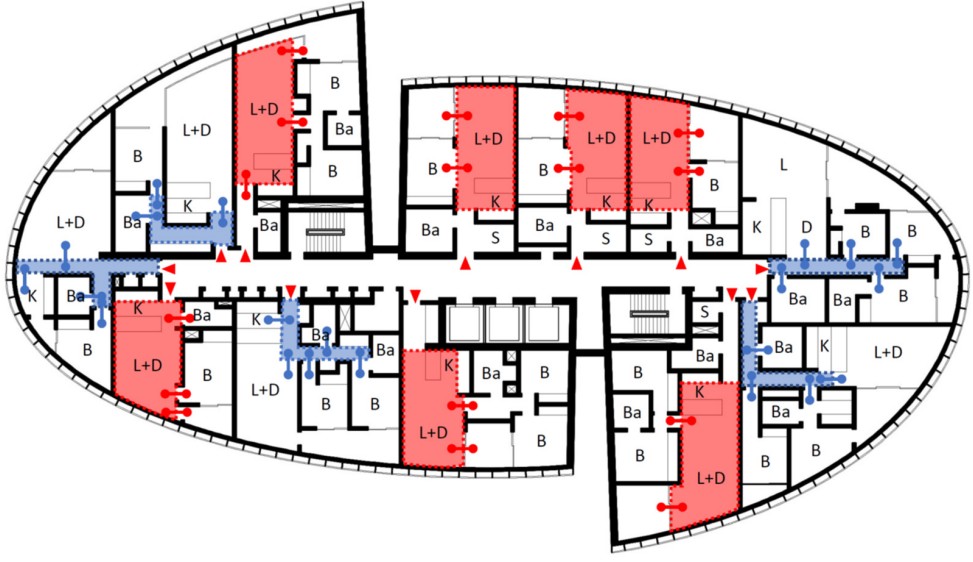

L: Living room; D: Dining room; K: Kitchen; B: Bedroom; Ba: Bathroom; S: Storage

Centralized hall or corridor    Centralized shared spaces

**Figure 10.** Unit types with a centralised living room and centralised hall or corridor in contemporary apartment buildings in Sydney. Australia Towers, Sydney (Sydney Olympic Park), designed by Bates Smarts, 2015 (Adapted with permission from Ref. [66]).

A centralised hall or corridor in modern apartments has a different function compared with the first and second eras (1900–1961). This is because they can facilitate separate spatial zones which encompass their own bedroom and bathroom (Figure 11). In this paper, we consider a separate zone as a spatially zoned single territory composed of bedrooms and bathrooms. This feature of in-unit layouts could have been influenced by demographic changes in apartment living at the time. For instance, there is a growing number of adult children living with their parents in Australia. According to a report by the Australian Institute of Health and Welfare [67], *"From 2007–2008 to 2017–2018, the proportion of young people aged 15–24 living with their parent/s (as a dependent student or non-dependent child) increased from 69% (or 2.0 million) to 75% (or 2.3 million). This increase was larger for 20–24 year olds (from 48% or 701,000 to 58% or 958,000) than for 15–19 year olds (from 91% or 1.3 million to 94% or 1.3 million)".* Given the fact that an increasing proportion of families with children live in units in Sydney (from 20.3 per cent to 25.1 per cent of apartment residents between 2006 and 2016) [26], this trend suggests a growing number of young adult children are living in units with parents who might benefit from a separate self-contained area with their own bedroom and bathroom. Considering the possibility of visiting relatives or independent children, retirees might also be more attracted by unit layouts facilitating an independent life achieved through such spatial zoning. Similar to the development of the "dual key system" for multigenerational households (for more detail, see [68]), a unit with two spatially separate zones with one containing its own bedroom and bathroom can be understood as a strategy to accommodate changing demographics. Another noticeable trend in the housing market in Sydney is the rise of share house living, which partly resulted from steeply rising housing unaffordability, not only for young adults but also for older adults [69]. Although share house living provides financial benefits, in particular for those with lower incomes, this can lead to poor living conditions. For example, such units generally have a shared bathroom used by three or more unrelated people [70]. A layout facilitating independent life through this spatial zoning of halls or corridors can, at least in part, resolve some of the challenges of shared housing living. Layouts of this present era are instructive in showing a home space that can be shared between nonfamily members. This is a notable feature compared with the layouts of the first era (1900–1935), in which sanitary facilities such as a common laundry, bathroom, and kitchen areas were sometimes shared inside the building but outside the unit itself.

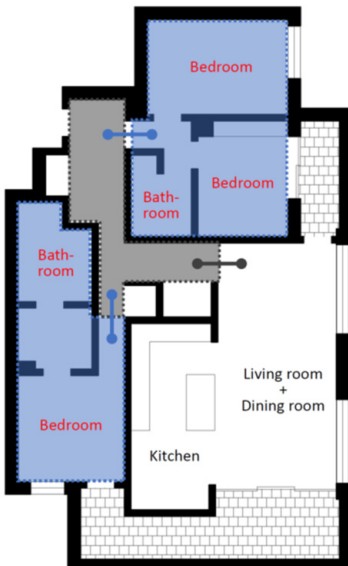

**Figure 11.** Layout for an independent life. Jacksons Landing, Sydney, 2003 (Adapted with permission from Ref. [71]).

Since 2002, substantive new regulations related to the apartment building and unit design have been introduced in the state of NSW. State Environmental Planning Policy 65 (SEPP 65) and the accompanying Residential Flat Design Code were introduced in 2002. SEPP 65 was the hardline legislation of the government against concerns about the proliferation of poor apartment design in NSW [72]. The Apartment Design Guide (ADG) [73], based on the Residential Flat Design Code, was developed and revised in 2015 in order to update the design quality of apartment buildings in NSW. It is used in conjunction with SEPP 65. The Building Sustainability Index (BASIX) was introduced in 2004 and mandated for the first time the minimum energy and water use in residential buildings in NSW.

Across these regulations, SEPP 65 and the ADG have had an especially significant influence on apartment layouts in Sydney because they provide minimum mandatory requirements for apartment building design. SEPP 65 aims to improve the design quality of apartment development in NSW [74], and the ADG provides benchmarks for designing and assessing these developments against SEPP 65 [73]. It has been suggested that SEPP 65 and the ADG, consequently, have improved the overall quality of apartment building designs, including spatial layouts [75–77], although some domestic scholars have expressed concerns that they are too prescriptive [75,78], which might limit innovation in apartment designs [75,76].

SEPP 65 and the ADG mandate that most habitable areas such as living rooms and bedrooms should be located on the perimeter of buildings providing access to light and ventilation. As a result, kitchens and bathrooms are generally located in the back of the units (Figure 12). For instance, the ADG requires that "*Living rooms and private open spaces of at least 70% of apartments in a building receive a minimum of 2 h direct sunlight between 9 am and 3 pm at mid-winter in the Sydney Metropolitan Area*" [73]. For acoustic privacy, it notes, "*Storage, circulation areas and non-habitable rooms should be located to buffer noise from external sources*"; hence many bathrooms and storage spaces are set adjacent to corridors [73].

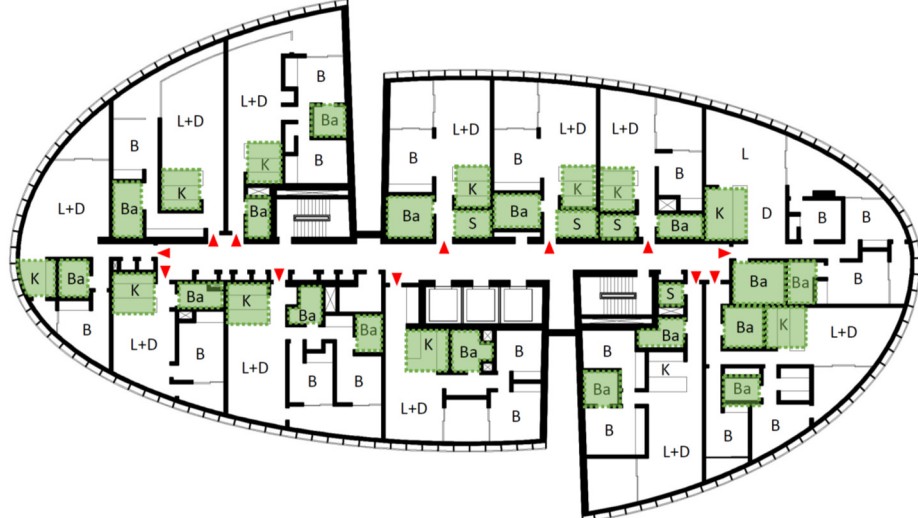

L: Living room; D: Dining room; K: Kitchen; B: Bedroom; Ba: Bathroom; S: Storage

Kitchens, bathrooms and storage in a unit

**Figure 12.** The location of kitchens, bathrooms, and storage in the units of contemporary apartment buildings in Sydney. Australia Towers, Sydney (Sydney Olympic Park), designed by Bates Smarts, 2015 (Adapted with permission from Ref. [66]).

In addition, many of the units constructed in this era are organised to promote cross ventilation for the improvement of indoor environmental quality. Specifically, the ADG provides the design criteria that "*At least 60% of apartments are naturally cross ventilated in the first nine storeys of the building. Apartments at ten storeys or greater are deemed to be cross*

*ventilated only if any enclosure of the balconies at these levels allows adequate natural ventilation and cannot be fully enclosed*" [73]. Figure 13 shows an example of a diagram showing cross ventilation for Development Application (DA) approval.

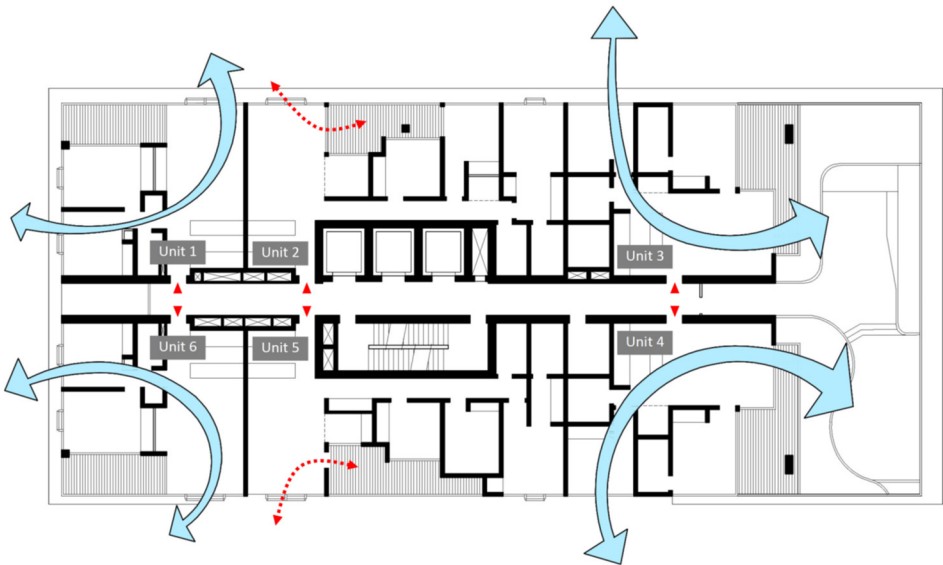

**Figure 13.** The apartment plan in Sydney designed for cross ventilation. In this layout, units 1, 3, 4, and 6 are considered cross-ventilated. Units 2 and 5 are single-side ventilated. In this case, 67% of units on this floor have access to cross ventilation. It is also worth noting that the living room space in units 2 and 5 has some access to cross ventilation because of the openings on perpendicular walls (highlighted in red) (Adapted with permission from Ref. [79]).

The spatial layout of units in this era, which are designed to promote daylight and cross ventilation to habitable rooms, have similarities to those of the first era. In both, the occupant's comfort, health, and amenity are drivers of the layout. Whereas in the first era, a lack of technological alternatives (artificial lighting and air-conditioning) meant the passive design was essential, in this era, a desire for energy efficiency and sustainability is more apparent.

At the building scale, apartment buildings generally have a layout of mixed units with different numbers of bedrooms within a single floor plate. Again, this is influenced by the ADG, which notes that in apartment design, "*A range of apartment types and sizes is provided to cater for different household types*" [73]. Figure 14 shows an example suggested by the ADG to deliver a variety of units in a single floor plate.

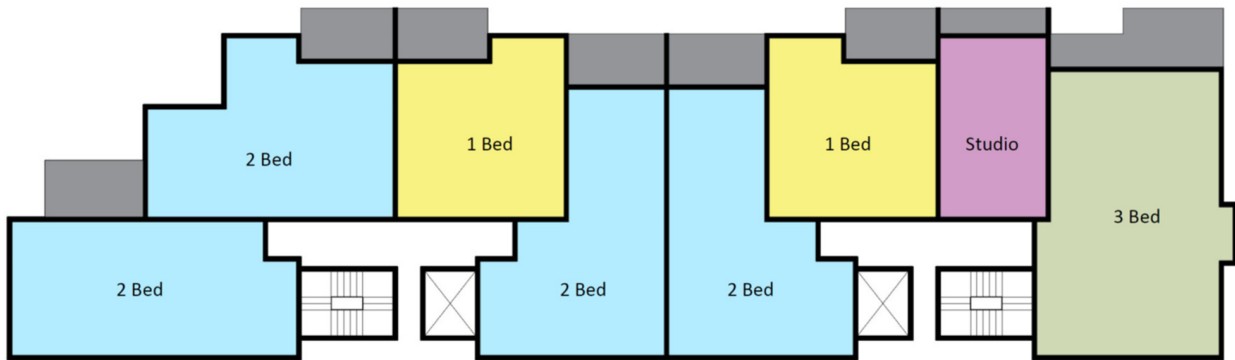

**Figure 14.** Building floor plan suggested by ADG for delivering a variety of units in a development (Adapted from Ref. [73]).

## 4. Findings

Through our historical analysis of the literature and apartment plans, we have identified four main categories of influence on apartment layouts: changes in (1) policy and regulation, (2) society and demography, (3) economy and the housing market, and (4) architectural paradigms and technologies. These are summarised in Figure 15, which provides a framework for understanding factors that directly or indirectly influenced unit and building layouts across social, economic, regulatory, and architectural spheres.

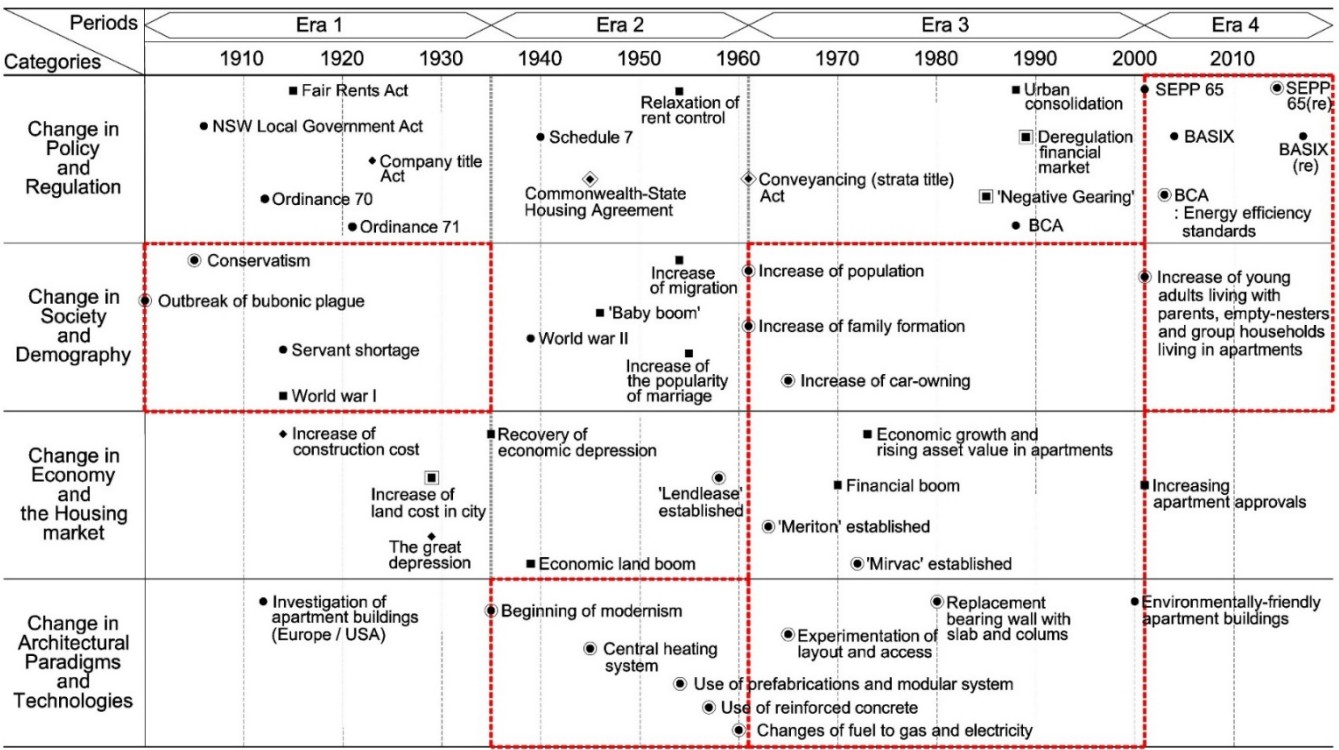

**Figure 15.** Analysis of historical events and developments that influenced apartment building design and spatial layouts in Sydney.

In all four eras examined, we found that events and factors in all these categories influenced apartment layouts at the time. However, in different eras, certain types of events or factors can be seen to have been more influential than others (as outlined with red-dotted boxes in Figure 15).

In comparing the features of spatial layouts in apartment buildings in Sydney and the main types of influences chronologically, we have identified four main eras of apartment layout:

- Layouts reflecting physically separate rooms and healthier living conditions: 1900–1935 (From the beginning of conservatism and the outbreak of plague);
- Layouts following function: 1935–1961 (From the growth of modernism and developments post World War II);
- Layouts enhancing interaction between family members: 1961–2002 (From a more significant growth in population as well as marriage rate, and the introduction of strata title);
- Layouts for independent life and satisfying minimum regulatory requirement: 2002–the present (From changes in demographic characteristics of households living in apartment buildings and the introduction of SEPP 65).

Social paradigm changes, such as the increase of conservatism [33] and safety issues from kitchen fires [34], were a significant influence on the compartmentalised layouts apparent before 1935. In addition, a desire to create healthy dwellings and tackle diseases facilitated layouts with lightwells, or where most rooms had access to light and ventilation. From 1935 to 1961, architectural paradigm changes, such as the growth of modernism [31] and technological developments such as central heating systems [19], drove efficient zoning by function and the elimination of open fireplaces. However, physically enclosed rooms for privacy and function still existed. These layouts changed between 1961 and 2002, driven again by social and demographic changes, including an increasing population as well as a change of occupants from renters to consumers (purchasers) of apartment buildings—change driven by economic and market changes and the emergence of major large developers. Over the same period, open-plan layouts emerged as the common spatial organisation of units, along with a period of innovation and experimentation in apartment building design. This was also facilitated by the widespread use of reinforced concrete structural systems, allowing open floor plans to become a reality, even at the height [55]. Since 2002, changes in society and demographics may have encouraged separate zoning for more independent lives to take place in a single unit while still maintaining open shared spaces for interaction. In addition, the emergence of design regulations influenced the standardization of apartments and sought to improve residential amenities through sunlight and ventilation in a comparable manner to the first era of apartment layout.

Table 1 summarises the overall findings of historical connections between the spatial layouts in apartment buildings in Sydney and the events and factors which are likely to have influenced them. The diagrams in Table 1 highlight the connectivity of particular spaces in order to visualise and compare the spatial layouts from different eras. The diagrams draw from space syntax theory [80] but differ from justified graphs and techniques because (1) physically open spaces were not divided by the concept of convex space; nonhabitable spaces such as open fireplaces and storage spaces are considered integral parts of unit planning, and (2) spatial hierarchy (depth) was not included. This was to simplify the characteristics of apartment layouts (the connectivity of particular spaces, including nonhabitable spaces such as open fireplaces and storage) by eras, rather than just the spatial characteristics (spatial hierarchy) of each room. In the diagrams in Table 1, solid line circles illustrate physically closed spaces by architectural elements such as walls, whereas dot-line circles show physically open spaces. Lines connecting circles signify spatial connectivity.

**Table 1.** The historical correlation between the spatial layouts in apartment buildings in Sydney and the influences.

| Eras | Influences (Events) | Features of Spatial Layout | |
|---|---|---|---|
| **1900–1935** (From the beginning of conservatism and the outbreak of plague to the growth of modernism and developments post World War II) | **International trends** **Changes in society** ○ Servant shortage ○ Conservatism ○ Outbreaks of disease and concern for residents' health and well-being | **Layouts reflecting physically separate rooms and a healthier living condition** [Unit layout] ○ Physically separate spaces for different functions ○ Centralised hall or corridor (which must be passed through in order to access other spaces beyond the unit entrance) ○ Open fireplaces ○ Zoning of living room, bedroom (or dining room), and open fireplace ○ The main bedroom directly faced the street [Building layout] ○ Communal service spaces (kitchen and laundry) ○ Most habitable rooms having access to windows or a lightwell for natural light and ventilation (resulting in undulating building plans) |  |
| **1935–1961** (From the growth of modernism and developments post World War II to more significant growth in population as well as marriage rate, and the introduction of strata title) | **Changes in architectural paradigm** ○ Growth of modernism **Development of technologies** ○ Central heating system by boiler and a shift to gas and electricity | **Layouts following function** [Unit layout] ○ Physically separate spaces for different functions ○ Centralised hall or corridor (through which all rooms were accessible) ○ Reduced open fireplaces ○ Spatial zoning of a kitchen and a bathroom [Building layout] ○ Common hall at the building entrance ○ Lightwells are no longer common |  |

Table 1. *Cont.*

| Eras | Influences (Events) | Features of Spatial Layout | |
|---|---|---|---|
| **1961–2002** (From a more significant growth in population as well as marriage rate, and the introduction of strata title to the changes in demographic characteristics of households living in apartment buildings and introduction of SEPP 65) | **Changes in demography**<br>○ Increase in population<br>○ Growth in marriage rate<br>**Changes in the economy and the housing market**<br>○ Recovery of economic conditions and rising asset value of apartments (from renters to owners)<br>○ Increase in land cost<br>○ Apartment development by large developers (such as Lend Lease, Mirvac, Meriton)<br>**Changes in architectural paradigm**<br>○ Split level apartments and open walkway access | **Layouts enhancing interaction between family members**<br>[Unit layout]<br>○ Centralized living room (which must be passed through in order to access other spaces beyond the unit entrance)<br>○ Open-plan living—i.e., LDK/L + DK<br>○ Standardisation by large developers<br>○ Increase in split level apartments<br><br>[Building layout]<br>○ Increase in single-loaded walkways and open-air access<br>○ Dedicated and enclosed parking areas in apartment buildings |  |
| **2002–the present** (From changes in demographic characteristics of households living in apartments and introduction of SEPP 65 to the present) | **Changes in society**<br>○ Housing unaffordability<br>**Changes in demography**<br>○ An increase of young adults living with parents, empty-nesters, and group households living in apartments<br>**Changes in society**<br>○ The State Environment Planning Policy 65 (SEPP 65)<br>○ Apartment Design Guide (ADG)<br>○ BASIX | **Layouts for independent life and satisfying minimum regulatory requirement**<br>[Unit layout]<br>○ Centralized hall, corridor, or living space (which must be passed through in order to access other spaces beyond the unit entrance)<br>○ Open-plan living, i.e., LDK/L + DK<br>○ The division of zoning for independent living areas<br>○ Living areas and bedrooms located on the perimeter of units<br>○ Provision of sunlight and ventilation to habitable rooms<br>○ Service areas (kitchens, bathrooms) located in the back of units<br><br>[Building layout]<br>○ Individual storage in common spaces<br>○ The organisation of units to better facilitate cross ventilation<br>○ Mixed unit sizes on floors |  |

H: Hall (or Corridor) in units/L: Living room/D: Dining room/K: Kitchen/B: Bedroom/Ba: Bathroom/F: Fireplace/E: Entrance/C: Hall (or Corridor) in buildings/St: Stair (or Lift)/P: Parking area/IS: Individual Storage/OS: Outdoor Spaces/G: Garbage disposal/CS: Communal spaces/S: Street; ○: Physically closed space/○: Physically open space.

## 5. Discussion: Impact on Future Apartment Layouts

This paper explored the historical evolution of apartment design in Sydney and provides a framework for understanding factors that directly or indirectly influenced unit and building layout across social, economic, regulatory, and architectural spheres. Changes in society and architectural paradigms and the development of technologies appear to be the dominant influence on apartment layouts before 1961, whereas changes in regulation and changes in the economy and housing market appear to have had a greater influence after 1961. The layouts of apartment buildings in Sydney seem to have been especially influenced by the introduction, and subsequent tightening, of mandatory design regulations since 2002. As noted, there have been few studies that examined factors influencing spatial layouts of apartments specifically. Thus, the framework we propose will support practitioners and academics in understanding how layouts have evolved over time and the issues and events that have informed this development and will continue to do so in the future. Spatial layout is an important attribute that influences the quality of life in apartments [6,7]. Therefore, we can conclude that the external factors identified here (outlined in Figure 15) are also subsequently impacting the lives and domestic occupations of the hundreds of thousands of people who reside within apartments in Sydney (and the millions who do so globally).

This thinking can also be extended to look into the future and how apartment layouts in Sydney (and elsewhere) might be influenced by significant socioeconomic and political events that we are facing now, most notably COVID-19 and the climate crisis. For instance, Aresta and Salingaros [81] criticize what they call the "deep malaise of contemporary architecture", captured in the way families were cooped up in minimalist internal spaces during COVID-19 lockdowns. In response, they propose the design of heterogeneous domestic spaces informed by human emotion as opposed to prescriptive spatial norms. Others have criticized the "functional" approach to internal circulation, typified by spatial layouts with central halls and boxy spaces on either side and instead promote more complex sequences of interconnected spaces [82]. During COVID-19 lockdowns, research has shown that some residents adapted by rearranging domestic furniture, but the physical transformation of rooms and layouts was limited in apartments. This suggests that more flexible and adaptable spatial layouts could be valuable in a post-COVID-19 world [83]. "Malleability", where occupants are able to respond to their changing needs and shape their own environment, is considered an important quality of spatial design [82]. Finally, there is a growing acknowledgement that access to outdoor space in dense urban areas is vital, and in apartments, the value of generous balcony spaces and even substantive sky gardens to support mental health and well-being and provide direct access to light and air, especially during pandemic events, is significant [4,84,85].

A growing body of research is also maturing on the importance of natural light and ventilation facilitating better indoor air quality in building design [86], and, in particular, apartment design [87–89], to mitigate the spread of disease and improve high-density living conditions. With increasing urban temperatures and heatwaves, there is also evidence that current apartment design paradigms in Australia and many warm climates globally would not sufficiently protect residents against heat stress in extreme weather events and that greater levels of insulation, reflective external surfaces, and increased ventilation, can be effective mitigation strategies [90]. In heatwaves in Sydney, it has been shown that urban temperatures can increase by up to 12 °C (compared with temperatures on a typical summer day), increasing the energy needed for building cooling and in heat-related morbidity and mortality [91].

In line with the fourth era of apartment layouts, it is likely that policy and regulation will be used to respond to some of these challenges. Most notably, a new Design and Place SEPP (State Environmental Planning Policy) was proposed in 2021 to subsume SEPP 65 and the Apartment Design Guide [92]. However, as of April 2022, the new NSW Planning Minister cancelled the Design and Place SEPP and has instead suggested an update of the BASIX standard. While its remit is broad, encompassing urban design, placemaking,

and environmental performance, many of the criteria in this regulation could influence future apartment design and layouts in Sydney. These include regulations to limit floor-plate sizes in towers, reduce the number of apartments in new buildings with single-sided ventilation, and promote greater cross ventilation strategies. It also proposes that apartments over ten stories high are no longer automatically deemed naturally ventilated, which will likely result in fewer compact apartment plans, and more fragmented apartment layouts with greater porosity and increased surface areas to facilitate cross ventilation. It is, therefore, likely that the fourth era, where regulation is the most significant influence on apartment layout in Sydney, will continue into the mid-term future. Given the historical example of the first era where regulations partially brought on by epidemics (such as Ordinances 70 and 71) influenced a healthier living environment in apartments, it is possible too we will see greater regulations governing indoor air quality in the wake of COVID-19. However, such regulatory influence needs to be carefully managed. As noted before, while there is a sense in the industry that regulations such as SEPP 65 and the Apartment Design Guide have lifted apartment outcomes in terms of design and amenity at the "bottom of the market", there is also concern that such a prescriptive approach has stifled innovation and creativity [75]. A potential solution is a shift to a more performance-based approach, where outcomes (such as thermal comfort, energy performance, or amenities) are regulated, but how these are achieved through the design and layout of apartments remains flexible for building practitioners and designers to achieve in diverse and innovative ways.

This research defines four eras of the spatial layout of apartments in Sydney. In the first era, apartments had physically distinct spaces for different functions, accessed from a centralised corridor with open fireplaces and lightwells. In the second, distinct spaces for different functions continue, though with more defined zoning for kitchens and bathrooms at the building scale and far fewer fireplaces and lightwells because of the rise of mechanical heating and lighting. The third era marked an increase in open-plan living, with the living space centralised, common split-level units, single-loaded walkways, and the inclusion of parking zones. The final (and current) era continues open-plan living but with a mix of the centralised hall and living spaces to zone units for different occupants and their lifestyles. To improve apartment layouts for residents, design patterns—rules that describe problems and provide potential design solutions—can be used [93]. While this research classifies apartment layouts historically, a greater understanding of design patterns that support rich spatial living experiences, health, well-being, and comfort in apartments can contribute to overcoming some of their spatial failings in the future and improve the daily lives of apartment-dwelling residents.

**Author Contributions:** Conceptualization, H.Y.; methodology, H.Y.; validation, H.Y.; formal analysis, H.Y., P.O. and H.E.; investigation, H.Y.; resources, H.Y.; data curation, H.Y.; writing—original draft preparation, H.Y.; writing—review and editing, H.Y., P.O. and H.E.; visualization, H.Y.; supervision, P.O. and H.E.; project administration, H.Y.; funding acquisition, H.Y. and P.O. All authors have read and agreed to the published version of the manuscript.

**Funding:** This research was funded by UNSW Sydney, University Postgraduate Award (UPA). The APC was funded by UNSW Sydney.

**Conflicts of Interest:** The authors declare no conflict of interest. The funders had no role in the design of the study; in the collection, analyses, or interpretation of data; in the writing of the manuscript, or in the decision to publish the results.

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
