# Peer review of "Influences on Apartment Design: A History of the Spatial Layout of Apartment Buildings in Sydney and Implications for the Future"

_buildings, doi:10.3390/buildings12050628_

Round 1

Reviewer 1 Report

The manuscript is well written and addresses an important topic that is not emphasized enough in research or literature. The topic of spatial layout of apartments and apartment buildings is quite interesting, not only from a research but also from a design point of view. The results of this and similar research are of great importance for the overall housing design practice, they can lead to the identification of various problems as well as offer the solutions. The manuscript provides an overview of changes in society, the economy, regulation and architectural paradigms, and shows how they have affected apartment layouts over time. Overall, the manuscript gives an overview of an interesting topic and is very informative and relevant, although there are two suggestions for its minor revision:

(1) Given the very long period observed in the research, it would be good to show the sample size (number of apartments and residential buildings) that were reviewed and on the basis of which the results were obtained. It is of great importance for the validity of the research to know whether the research was done only on apartments and buildings shown through the blueprints in the manuscript or the number is much higher, both for the entire research and for each of the defined periods. Please complete chapter 2. Materials and Methods with sample size information.

(2) Results review and chapter 4. Findings are very well and thoroughly presented and described both through text and through tables and floor plans, but in chapter 5. Discussion and Conclusion, there is very little discussion and emphasis on the most important conclusions of your research. It is correct and justified to analyze and question the results of research in the context of future events regarding housing in Sydney, but it should not prevail in the conclusion. You should focus on your own research and highlight its findings.

Reviewer 2 Report

The research is presented under the pretense that there is no publication on the events that influenced the layout of Sydney apartments in the 20th c, while the authors ignore the contents of Butler-Bowden's dissertation (2009) on the same topic and also an earlier publication on Australian apartments. Although both publications are mentioned in the references (30 and 17). Butler-Bowden probably also uses eras to describe the changes, however I could not get hold of this dissertation to check if she does. The only potential of the authors' research could be to point out what is missing in Butler's approach, such as more recent developments.  

Presenting first the eras as 'results' of the method is confusing and not instructive, for it is unclear what determined the periodization. 1961 seems an important year, like 1935 but why? The layout (fig.2a) to exemplify the era 1900-1935 presents without explanation an apartment without kitchen and dining room. Probably an apartment hotel with collective kitchen and dining room. The layouts to exemplify the 1935-1961 era are all post-war. Functionalism in the domestic domain concentrated in the 1920s and 30s on the kitchen and the connection with the living/dining room. The layouts of the present era are most instructive in showing the sharing of intimate domestic space by non-family members, while the layouts of the first era pointed to the sharing of facilities like kitchen and dining outside the domestic domain.  The first era was determined by the servant shortage and the recent era by a housing shortage and sky high real estate prices.

Reviewer 3 Report

The work is very interesting for me personally, but unfortunately after the second reading I felt that something was missing. After the third review of the work and the suggested correction, I saw an error in my judgment. I miss photos or drawings of the facade that would connect to the floor plans. I do not claim that they are necessary for this paper, but looking at the period analyzed and the changes in floor heights  that followed over the decade, I think that this plays a role, especially in terms of air volume and thermal comfort provided by the building.

Now a serious criticism from the engineer in me: footnotes in scientific papers are characteristic of professions that do not have a strong connection to engineering, leave them out of the paper because they serve no purpose. 

I suggest splitting Figure 1 into two different figures. Although the word percent is there, the unit symbol (%) should be written. Source: [26], is the author's graph not a logical spelling, because it is quite clear that the authors wrote the numbers in Excel, mainly because of the strange symbols in the 'a)' part of the figure?

On all images there is the note 'drawing redrawn by author'. Please omit this note or put it only once at the beginning of the article. I am not sure if this note is necessary at all, because due to the homogeneity of the approach and the fact that there was no CAD a hundred years ago, it is clear that the authors drew the figures.

Round 2

Reviewer 2 Report

My main criticism has been addressed, and I agree to the publication.
